# Physical Activity, Stress, Depression, Emotional Intelligence, Logical Thinking, and Overall Health in a Large Lithuanian from October 2019 to June 2020: Age and Gender Differences Adult Sample

**DOI:** 10.3390/ijerph182312809

**Published:** 2021-12-04

**Authors:** Albertas Skurvydas, Ausra Lisinskiene, Marc Lochbaum, Daiva Majauskiene, Dovile Valanciene, Ruta Dadeliene, Natalja Fatkulina, Asta Sarkauskiene

**Affiliations:** 1Education Academy, Vytautas Magnus University, K. Donelaičio Street 58, 44248 Kaunas, Lithuania; albertas.skurvydas@vdu.lt (A.S.); marc.lochbaum@ttu.edu (M.L.); daiva.majauskiene@vdu.lt (D.M.); dovile.valanciene@gmail.com (D.V.); 2Department of Rehabilitation, Physical and Sports Medicine, Faculty of Medicine, Institute of Health Sciences, Vilnius University, 21/27 M.K. Čiurlionio St., 03101 Vilnius, Lithuania; ruta.dadeliene@mf.vu.lt; 3Institute of Educational Research, Education Academy, Vytautas Magnus University, K. Donelaičio Street 58, 44248 Kaunas, Lithuania; 4Department of Kinesiology and Sport Management, Texas Tech University, Lubbock, TX 79409-3011, USA; 5Department of Physical and Social Education, Lithuanian Sports University, Sporto Street 6, 44221 Kaunas, Lithuania; 6Institute of Sport Science and Innovations, Lithuanian Sports University, Sporto Street 6, 44221 Kaunas, Lithuania; 7Institute of Health Sciences, Faculty of Medicine, Vilnius University, 21/27 M.K. Čiurlionio Street, 03101 Vilnius, Lithuania; natalja.fatkulina@mf.vu.lt; 8Department of Sports, Recreation and Tourism, Klaipėda University, Herkaus Manto Street 84, 92294 Klaipėda, Lithuania; asta.sarkauskiene@ku.lt

**Keywords:** moderate-to-vigorous physical activity, physical health perceptions, mental health, cognitive functioning, personality

## Abstract

This study aimed to examine relationships and group differences among adult people’s (aged 18–74) physical activity (PA), expression of stress, depression, emotional intelligence (EI), logical thinking (LT), and overall health assessment. Two hypotheses were formulated before the study. The first hypothesis is that overweight and obesity in young adults (18 to 34 years) females and males, in particular, should increase sharply and this should be associated with decreased PA, abruptly deteriorating subject health, increased stress, depression, and poorer emotion management and EI. Second hypothesis: We further thought that the better people’s reflective thinking, the more they should live a healthier life (e.g., exercise more and eat healthier), their overweight and obesity should be small or none. We aimed to confirm or reject these two hypotheses. We applied a quantitative cross-sectional study design. The study results revealed that during the lifespan of 18–24 and 25–34 years (young adults) there was a sharp increase in overweight and obesity, a decrease in PA (and especially vigorous physical activity (VPA)) (and this was particularly evident in the male), while research participants felt less stress and depression, subjective assessment of health did not change, and EI increased steadily with age (18–24 to 65–74 years). The higher the EI of the research participants from 18–24 to 65–74 years of age the higher their moderate-to-vigorous physical activity (MVPA), the less stress and depression they felt. Based on the results, it can be said that both females and males prefer PA “with a hot heart rather than a cold mind.” We base this conclusion on the fact that females and males who have the highest EI also have the highest MVPA while LT is not associated with MVPA.

## 1. Introduction

There is growing evidence-based research that various forms and doses of PA are effective in combating many chronic diseases [1,2,3], improving well-being and mental health [4,5,6], improving emotional well-being [7], and reducing all-cause mortality [8,9]. It was found that the effect of PA on various body functions is quite specific (i.e., depends in a non-linear manner on muscle work intensity, duration, and load “doses”) [1,3,10]. In addition, the health benefits of PA also depend on people’s age, gender, health, and body mass index (BMI) [3,4,8,11,12]. However, the lack of PA can negatively affect people’s wellbeing and their overall health. The Word Health Organization (WHO) recommendations highlight that adult people must be physically active at least 150 min per week [3]. However recent studies have shown [3] that worldwide adult people do not meet WHO recommendations.

One recent meta-analysis showed a rising trend in the overall prevalence of obesity since the 1990s and this trend was more drastic in young adult female and male individuals [13]. It has been clearly shown that lack of PA increases the incidence and subsequently causes systemic inflammation of the body, which causes many chronic diseases [1,14,15]. Moreover, depressive symptoms are often associated with obesity, and emotional eating may play a considerable role in weight gain [16]. In addition, [17] have found that emotional eating (i.e., eating in response to negative emotions) is one mechanism linking depression and subsequent development of obesity. However, studies have rarely examined this mediation effect in a prospective setting and its dependence on other factors linked to stress and its management. The researchers [17] concluded that emotional eating is one behavioral mechanism between depression and development of obesity and abdominal obesity. Moreover, adults with a combination of shorter night sleep duration and higher emotional eating may be particularly vulnerable to weight gain. In relation to this, researchers [18] found that stress awareness was significantly associated with weight gain among women, while other psychological factors were not significantly associated with weight gain. Among women with overweight and obesity, stress is the main variable associated with excessive gestational weight gain [19].

Based on the [20] results, there is an urgent need for local and culturally contextualized interventions to address the complex risk factors that impact being overweight and obese, including the nutrition of at-home cooking, specific physical activity barriers, such as finding time, and perceptions of healthy weight for reducing the risk of long term cardiovascular disease. The researchers [21] summarized the dynamics of the 1.9 million human physical inactivities from 2001 to 2016: In developed European countries, it increased significantly for both males and females. Obesity and low PA can be intertwined (i.e., low PA stimulates obesity, while obesity has reduced people’s motivation to be PA [22], and stimulates more frequent overeating due to the inability to suppress eating cravings [23]). Evidence from epidemiological studies indicates that depression and obesity have a strong bidirectional relationship (i.e., BMI increases the risk for developing depression, and vice versa, individuals with depression have an increased risk of high BMI) [24]. Stress and negative emotions during childhood pose a major threat to public health since they have been related not only to psychological disease but also to physiological disturbances such as obesity [25].

Based on the findings of the studies cited above, two hypotheses have been highlighted forward. The first hypothesis: Overweight and obesity in young adults—from 18 to 34 years—females and males, in particular, should increase sharply and this should be associated with decreased PA, abruptly deteriorating subject health, increased stress, depression, and poorer emotion management/EI. Second hypothesis: Since human choices are not necessarily influenced by a rational/reflective mind (explicit knowledge), but it often depends on implicit knowledge (i.e., from EI, impulsivity [26,27]), we further thought that the better people’s reflective thinking, the more they should live a healthier life (e.g., exercise more and eat healthier) (i.e., their overweight and obesity should be small or none). The study aims to confirm or reject these two hypotheses.

This research to our knowledge is the only one which presents the Lithuanian adult people sample in relation to PA, weight and obesity, expression of stress and depression as well as the health assessment. It presents the overall picture of Lithuanian situation regarding PA in adult people context.

## 2. Materials and Methods

### 2.1. Participants

The participants were 6369 research participants (females = 4545 and males = 1824) between the ages of 18- and 74-year-old (Figure 1) completed correctly. The investigations were conducted from October 2019 to June 2020. The participants were included from Lithuania country to represent the Lithuanian sample. Participation was anonymous. Thus, data collection and handling were confidential. We used an online survey to collect information through https://docs.google.com/forms/ (accessed on 1 March 2020). All participants completed the online questionnaires. An online survey using the Google Forms platform.

### 2.2. Procedure

Ethic committee approval to conduct this research was provided by the Klaipeda University (protocol No. STIMC-BTMEK-08). Besides, we ensured that the study was performed according to the principles laid by declaration of Helsinki [28] and National guidelines for biomedical and health research involving human participants [29]. The purpose of the survey, introduction and about the length of the survey was added within the web-based open E-survey. A successful return of completed survey was considered as consent by the participant.

### 2.3. Measures

Danish Physical Activity Questionnaire (DPAQ). The DPAQ is adapted from the IPAQ instrument and differs from this by referring to PA of the last 24 h (for seven consecutive days) instead of the last 7 days [30] in metabolic equivalent (METs). One metabolic equivalent (MET) is defined as the amount of oxygen consumed while sitting at rest and is equal to 3.5 mL O2 per kg body weight × in. The chosen activities were listed in the PA scale in nine levels of physical exertion, ranging from sleep or inactivity (0.9 MET) to strenuous activities (>6 METs). Each level (A = 0.9 MET, B = 1.0 MET, C = 1.5 METs, D = 2.0 METs, E = 3.0 METs, F = 4.0 METs, G = 5.0 METs, H = 6.0 METs and I > 6 METs) was described by examples of specific activities of that particular MET level and by a small drawing.

The PA scale was constructed so that the number of minutes (15, 30, or 45 min) and hours (1–10 h) spent on each MET activity level on an average 24 h weekday could be filled out. This allowed for a calculation of the total MET-time, representing 24 h of sleep, work, and leisure time on an average weekday [31].

It has been calculated how much energy (METs) was consumed per day during sleeping, sedentary behavior (SB) (from 0.9 to 1.5 Mets), light-intensity physical activity (LPA) (from 1.5 to 5 METs), moderate-intensity physical activity (MPA) (from 3 to 6 METs), vigorous-intensity physical activity (VPA) (more than 6 METs). Besides, we have combined MPA and VPA (MVPA).

Subjective health assessment. A four-point scale was used for this: poor health (1 point); satisfactory (2 points), good (3 points); excellent (4 points).

Perceived stress and depression. The 10-item perceived stress scale (PSS-10) was used to measure participants’ stress levels [32]. In the PSS-10, participants were asked to answer 10 questions about feelings and thoughts during the last month on Likert scale ranging from 0 (never) to 4 (very often), indicating how often they have felt or through a certain way within the past month. Scores range from 0 to 4, higher scores indicate higher levels of perceived stress.

Assessment of emotional intelligence. EI was assessed using the Schutte self-report emotional intelligence test (SSREIT) [33]. The SSREIT is a 33-item questionnaire divided into four subscales: perception of emotion assessed by 10 items, managing own emotions assessed by 9 items, managing others’ emotions assessed by 8 items, and utilization of emotions assessed by 5 items. The items are answered on a five-point scale ranging from 1 (strongly agree) to 5 (strongly agree). Total scores range from 33 to 165, with the higher scores indicating greater ability in EI.

Assessment of impulsivity. Impulsivity was assessed using the Barratt impulsiveness scale version 11 (BIS-11) [34]. The BIS-11 is a 30-item questionnaire divided into three subscales: Attentional impulsiveness assessed by 8 items, motor impulsiveness assessed by 11 items, and non-planning impulsiveness assessed by 11 items. The items are answered on a four-point scale ranging from 1 (rarely/never) to 4 (almost always/always). Total scores range from 30 to 120, with higher scores representing higher impulsivity.

The Cognitive Reflection Test (CRT). The test tasks were developed according to the CRT test discussed in the article by [35]. The test consists of three tasks, which, after reading, automatically select the wrong answer. The author notes that it is possible to check what kind of thinking system a person uses. The first system reflects intuitive decision-making, which is usually fast, automatic, requires minimal effort, is implicit, and emotional. Meanwhile, the second system reflects reasoning that is slower, conscious, effort-intensive, goal-oriented, and logical. The test consists of three questions, for example: (1) A bat and a ball cost $1.10 in total. The bat costs $1.00 more than the ball. How much does the ball cost? _____ cents (2) If it takes 5 machines 5 min to make 5 widgets, how long would it take 100 machines to make 100 widgets? _____ minutes (3) In a lake, there is a patch of lily pads. Every day, the patch doubles in size. If it takes 48 days for the patch to cover the entire lake, how long would it take for the patch to cover half of the lake? _____ days. The measure is scored as the total number of correct answers. The cognitive reflection test (CRT) measures cognitive processing—specifically the tendency to suppress an incorrect, intuitive answer and come to a more deliberate, correct answer.

### 2.4. Data Analysis

Interval data are reported as the mean ± standard error. All interval data were confirmed as being normally distributed using the Kolmogorov–Smirnov test. One-way and two-way ANOVA were performed to assess the effect of independent variables on the dependent variables of MVPA (METs). Calculations for observed power (OP) were performed; the partial eta squared (${ŋ}_{P}^{2}$) value was estimated as a measure of the effect size, and the B coefficient was estimated as a parameter of the regression. Chi-square (χ^2^) tests were conducted to compare differences across sexes. The relationship between variables was also assessed with Pearson’s correlation coefficient. For all tests, statistical significance was defined as *p* < 0.05. Statistical analyses were performed using IBM SPSS Statistics software (version 22; IBM Corp., Armonk, NY, USA).

## 3. Results

### 3.1. Health, Logic Task Solution, Impulsivity and EI: Differences between Gender

Female and male values concerning health, logical task solution, impulsivity (BIS), and EI are described in Table 1. It has been found that men were statistically significantly better at assessing their health, solving logical problems better than women (Table 1). However, women had a statistically significantly higher EI. Interestingly, men and women did not differ in impulsivity.

### 3.2. PA Determinants: Male vs. Female

The results of the study showed that total METS per day was higher in male than in female, with age increasing (effect of Age: *p* < 0.0001; ${ŋ}_{P}^{2}\;$= 0.026; OP = 1; effect of Gender: *p* < 0.0001; ${ŋ}_{P}^{2}$ = 0.007; OP = 1; Age × Gender: *p* = 0.018; ${ŋ}_{P}^{2}\;$= 0.002; OP = 0.82) (Figure 2A and Figure 3).

It was found that younger adults (18–24 years of age) sleep more compared to adults 55–64 years of age (*p* < 0.05) only for both female and male and there was no difference between males and females (effect of Age: *p* < 0.0001; ${ŋ}_{P}^{2}\;$= 0.011; OP = 1; effect of Gender: *p* = 0.134; descriptive average values in female and 0.000; OP = 0.31; Age × Gender: *p* = 0.82; ${ŋ}_{P\;}^{2}$= 0.000; OP = 0.16) (Figure 2B and Figure 3).

The results showed that METs of SB adults of 45–54 years of age SB is higher compared to adults of 65–74 years of age (*p* < 0.05) and there was no difference between male and female (effect of Age: *p* < 0.0001; ${ŋ}_{P\;}^{2}\;$= 0.014; OP = 1; effect of Gender: *p* = 0.11; ${ŋ}_{P}^{2\;}$= 0.000; OP = 0.35; Age × Gender: *p* = 0.25; ${ŋ}_{P}^{2}\;$= 0.001; OP = 0.476 (Figure 2C and Figure 3).

LPA METs for female of all ages were higher than for male and increased with age for both male and female (effect of Age: *p* < 0.0001; ${ŋ}_{P}^{2}\;$= 0.007; OP = 1; effect of Gender: *p* < 0.0001; ${ŋ}_{P}^{2\;\;}$= 0.015; OP = 1; Age × Gender: *p* = 0.074; ${ŋ}_{P}^{2}\;$= 0.002; OP = 0.68 (Figure 2D and Figure 3).

It is interesting to note that METs of MPA regarding age there was no difference between male and female (effect of Age: *p* < 0.0001; ${ŋ}_{P\;}^{2}$= 0.016; OP = 1; effect of Gender: *p* = 0.13; ${ŋ}_{P}^{2\;}$= 0.000; OP = 0.35; Age × Gender: *p* = 0.062; ${ŋ}_{P}^{2}\;$= 0.002; OP = 0.71) (Figure 2E and Figure 3).

Research results showed that male (18–24 years of age) METs of VPA was higher than female from 18–24 years (effect of Age: *p* < 0.0001; ${ŋ}_{P}^{2}\;$= 0.017; OP = 1; effect of Gender: *p* < 0.0001; ${ŋ}_{P}^{2}$= 0.029; OP = 1; Age × Gender: *p* = 0.287; ${ŋ}_{P}^{2}\;$= 0.001; OP = 0.44) (Figure 2F and Figure 3).

Research results showed that METs of MVPA were higher in male than in female and varied significantly with age (effect of Age: *p* < 0.0001; ${ŋ}_{P}^{2}\;$= 0.011; OP = 1; effect Gender: *p* < 0.0001; ${ŋ}_{P}^{2}\;$= 0.061; OP = 1; Age × Gender: *p* = 0.059; ${ŋ}_{P}^{2\;}$= 0.002; OP = 0.708 (Figure 2G and Figure 3). Interestingly, for male and female aged 18–24 to 35–44 years, it decreased significantly and increased from 35–44 to 65–74 years (*p* < 0.05).

There are more non-athletic females than males (*p* <0.05) (Table 2).

### 3.3. Participation in Sports

From 18–24 to 25–34 years, the number of non-athletes females was significantly low (*p* < 0.05) compared to independently exercising females (*p* < 0.05) (Table 2). Males prefer to do sports compared to females on their own (independently), but females prefer to play sports in sports clubs. The effect of age on participation in sports was significant in males and females (chi-square 101.6 and 169.9, respectively, *p* < 0.0001 for both females and males).

### 3.4. Changes in BMI during Age in Female and Male

BMI significantly is higher of 55–64 years of adults compared to 18–24 years of age and continued to stabilize in both male and female (effect of Age: *p* < 0.0001; ${ŋ}_{P}^{2}\;$= 0.066; OP = 1) (Figure 4). BMI was higher in males (except 65–74 years) than in females of all ages (effect of Gender: *p* = <0.0001; ${ŋ}_{P}^{2}\;$= 0.011; OP = 1). With age, normal BMI and low BMI (<20 kg/m^2^) decreased in males and females, overweight and obesity increased (chi-square female age effect 512.6, *p* < 0.0001; male—203.6, *p* < 0.0001) (Table 3). Besides, females with BMI were <20 kg/m^2^, there was more than males (*p* < 0.0001). Interestingly, for males with a BMI of 25–29.9 kg/m^2^, it increased from 23.9 to 42.5%, respectively, from 18–24 to 25–34, compared with 11 to 16.3% among females (Table 3).

### 3.5. Changes in Health, EI, Depression, Impulsivity, Perceived Stress and Reflexive Thinking (Logic Task Solution) with Age

Research results showed that male’s health scores are better than female’s (effect of Gender: *p* = <0.0001; ${ŋ}_{P}^{2}\;$= 0.03; OP = 0.99), but for both male and female it decreased significantly since 55–64 years (effect of Age: *p* < 0.0001; ${ŋ}_{P\;}^{2}$= 0.011; OP = 1; Age × Gender: *p* = 0.42; ${ŋ}_{P}^{2}\;$= 0.001; OP = 0.36) (Figure 5A).

Interestingly, EI was higher in female of all ages than in male (*p* = <0.0001; ${ŋ}_{P}^{2}\;$= 0.004; OP = 0.99), however, for both female and male, it increased significantly from 18–24 to 55–64 years (effect of Age: *p* < 0.0001; ${ŋ}_{P}^{2}\;$= 0.014; OP = 1; Age × Gender: *p* = 0.96; ${ŋ}_{P}^{2}$ = 0.000; OP = 0.096) (Figure 5B).

Depression symptoms was lower in male than in female (effect of Age: *p* < 0.0001; ${ŋ}_{P}^{2\;}$= 0.023; OP = 1; effect of Gender: *p* < 0.0001; ${ŋ}_{P}^{2\;}$= 0.03; OP = 0.99; Age × Gender: *p* = 0.52; ${ŋ}_{P}^{2\;}$= 0.001; OP = 0.31) (Figure 5C).

Interesting to note that the impulsivity (BIS) did not depend on age or gender (effect of Age: *p* = 0.11; ${ŋ}_{P}^{2}\;$= 0.011; OP = 1; Gender: *p* = 0.741; ${ŋ}_{P}^{2}\;$= 0.0000; OP = 0.063. Age × Gender: *p* = 0.74; ${ŋ}_{P\;}^{2}$= 0.008; OP = 0.21) (Figure 5D).

Research results showed that perceived stress was higher in adults of 18–24 years of age compared to 45–54 years of age adults (effect of Age: *p* < 0.0001; ${ŋ}_{P}^{2}\;$= 0.02; OP = 1). Perceived stress in all female group were higher (effect of Gender: *p* < 0.0001; ${ŋ}_{P\;}^{2}$ = 0.003; OP = 0.98; Age × Gender: *p* = 0.96; ${ŋ}_{P\;}^{2}$ = 0.000; OP = 0.09) (Figure 5E).

Besides, it has been found that reflective thinking (logic task solution) has an impact regarding adult age: Younger adults’ reflective thinking is higher than older adult people (*p* = 0.0041; ${ŋ}_{P}^{2}\;$= 0.003; OP = 0.91), males scored higher (effect of Gender: *p* = 0.008; ${ŋ}_{P}^{2}\;$= 0.000; OP = 0.75; Age × Gender: *p* = 0.79; ${ŋ}_{P}^{2}\;$= 0.000; OP = 0.18) (Figure 5F).

### 3.6. The Effect of BMI on Perceived Stress, Health and Depression

Interestingly, perceived stress and depression in female was independent of BMI (Figure 6). Quite unexpected that even with a higher BMI in males, the lower the perceived stress and depression (*p* < 0.05) (Figure 6A,C). In addition, a health assessment is, thus, independent of BMI (Figure 6B).

### 3.7. The EI on Perceived Stress, MVPA, Depression and Health in Different Ages

Research results showed that effect of EI on perceived stress is significant (*p* < 0.0001; ${ŋ}_{P}^{2}\;$= 0.018; OP = 1; Age × EI: *p* = 0.85; ${ŋ}_{P}^{2}\;$= 0.001; OP = 0.46) (Figure 7A).

Besides, the higher the EI, the higher the score of MVPA (*p* < 0.0001; ${ŋ}_{P}^{2}\;$= 0.006; OP = 1; Age x EI: *p* = 0.91; ${ŋ}_{P}^{2}\;$= 0.001; OP = 0.37) (Figure 7B).

Interestingly, the higher the EI, the lower all age groups’ depression level (*p* < 0.0001; ${ŋ}_{P}^{2}\;$= 0.014; OP = 1; Age x EI: *p* = 0.72; ${ŋ}_{P\;}^{2}$= 0.002; OP = 0.53) (Figure 7C).

We found that health condition did not depend on EI (*p* = 0.22; ${ŋ}_{P}^{2}\;$= 0.001; OP = 0.31); however, it was found that in relation to people 55–64 years of age, the higher the EI they reported, the higher and better health condition they stated (*p* < 0.05) (Figure 7D).

### 3.8. The Effect of Reflective Thinking on MVPA and Depression during Different Ages

The results of the study showed that MVPA of people of different ages does not depend on reflective thinking (logic solution task) (effect of reflective thinking: *p* = 0.74; ${ŋ}_{P}^{2}\;$= 0.000; OP = 0.13; Age × reflective thinking: *p* = 0.81; ${ŋ}_{P}^{2\;}$= 0.002; OP = 0.465) (Figure 8A). Besides, effect of reflective thinking on depression was also not significant (*p* = 0.56; ${ŋ}_{P\;}^{2}$= 0.000; OP = 0.21; Age × reflective thinking: *p* = 0.59; ${ŋ}_{P}^{2}\;$= 0.002; OP = 0.61). (Figure 8B).

## 4. Discussion

This study aimed to examine Lithuanian adult people’s (aged 18–74) PA, EI, expression of stress, depression, and their overall health status. 

Study results showed that the structure of PA is different in males and females: Males have a higher VPA and females have a higher LPA (Figure 2 and Figure 3). Males under 55 years old have a higher total daily energy expenditure and METs of MVPA than females. Higher PA in males is associated with more males exercising more independently.

Moreover, the BMI of a male of all ages (except 65–74 years) is higher than that of a female (Figure 4). From 18–24 to 55–64 years, BMI increased for both males and females. The number of males who are overweight increased sharply from 18–24 to 25–34 years (Table 3) and this is associated with a sharp decrease in daily energy expenditure in that age group (Figure 2 and Figure 3).

In addition, health assessment (for both male and female) did not change from 18–24 to 45–54 (Figure 5A), while that between depression (Figure 5C) and perceived stress (Figure 5E) decreased steadily at that stage, with the decrease thought to be significantly influenced by the increase in EI with age (Figure 5B) (impulsivity did not change with reflective thinking decreased during life-span). Depression (Figure 7C) and perceived stress (Figure 7A) were the highest and MVPA (Figure 7B) the highest among people of all ages with the lowest EI. Interestingly, people of all ages with the highest EI had the same health outcomes and did not change during their lifespan (Figure 7D). Interestingly, health assessment did not depend on BMI (Figure 6B), and males with higher BMI even felt less stress (Figure 6A).

In general, our primary hypothesis was only partially confirmed then, as we expected, overweight and obesity in young adults—from 18 to 34 years—increased and especially in the male, and at the same time the lifespan interval decreased in MVPA (consistent with data from other researchers, [11]. Contrary to expectations, subjective health assessment did not change, and feelings of stress and depression decreased (for impulsiveness did not change with age). In addition, EI increased at that age and up to 74 years. We believe that the increase in EI with age has been gelled to “compensate” for the negative effects of an increase in BMI on subjective health assessment, less depression, and stress. People are experiencing a sudden increase in overweight and obesity, but EI (and of course other reasons) does not allow us to state it as “bad health”. This is consistent with data from other researchers that various social and psychological factors have a positive effect on people’s well-being over their lifespan [36]. In addition, during one’s lifespan, different life goals and motives for achieving them appear. For example, between 18–24 and 25–34, both males and females face their first job, marriage, and the birth of a child [37], which may contribute to a decrease in MVPA. Besides, the PA of humans is influenced by several interrelated determinants such as demographic, health and health behavior, psychological, social, environmental, and determinants related to the intervention [12,38,39]. In addition, studies by Stone et al., 2017 show that Americans sense of stress increased from 20 to 50 years of age and began to decrease from 50 years of age. However, in our case, the stress started to decrease steadily from the age of 18–24 to 65–74.

Research results completely reject the second hypothesis, where increasing BMI with age, reflective thinking even decreased. Interestingly, the higher the BMI in males, the lower the stress and depression (Figure 6A,B). In our case, with a sharp increase in BMI from 18–24 to 25–44 years and a decrease in PA, it would appear that people are about to make an explicit decision to combat the severity of a very important chronic disease (obesity) [40]. Therefore, we can only speculate that people decided to do PA or not based more on implicit knowledge rather than explicit (reflective thinking), because implicit thinking requires less effort, it is more psychologically attractive [26,27]. Research by [41] has shown that people who cope well with stress and have an increased BMI have better cardiovascular health than those who do not manage stress and have a healthy BMI. The fact that our research findings show that females felt less depressed than males coincides with the work of other researchers [42]. Therefore, it can be said that both females and males prefer PA “with a hot heart rather than a cold mind.” We base this conclusion on the fact that females and males who have the highest EI also have the highest MVPA, although LT is not associated with MVPA. This conclusion of our study is related to the findings of other researchers that EI is related to the level of human health [43,44], PA [7], and rapid decision-making [45]. Based on the cognitive reflection assessment [36], we found that, in contrast to EI, cognitive reflection is not associated with MVPA.

## 5. Limitations and Directions for Future Research

This study examined adult people’s structure of PA, stress, LT, EI, and overall health in Lithuanian people. To estimate the observational processes mentioned above aspects, longitudinal studies are particularly urgent. Moreover, this study focused on European adult people (Lithuanian), thus the cultural differences about a similar topic are also particularly needed. In addition, future studies are highly needed to include the adolescent stage of age observations in a similar type of study.

## 6. Conclusions

This research study clearly showed that during the lifespan of 18–24 and 25–34 years (young adults) there was a sharp increase in overweight and obesity, a decrease in PA (and especially vigorous PA; and this was particularly evident in a male), while research participants felt less stress and depression, subjective assessment of health did not change, and EI increased steadily with age (18–24 to 65–74 years). The higher EI of the research participants—18–24 to 65–74 years of age—the higher their moderate-to-vigorous PA and the less stress and depression they felt. It was found that “vigorous PA did not depend on reflective thinking”. Thus, EI helps one to feel better, but “masks” a huge scourge of health deterioration—overweight and obesity and physical inactivity. Therefore, it can be said that both females and males prefer PA “with a hot heart rather than a cold mind.” This conclusion is based on the fact that females and males who have the highest EI also have the highest MVPA, although LT is not associated with MVPA.

## Figures and Tables

**Figure 1 ijerph-18-12809-f001:**
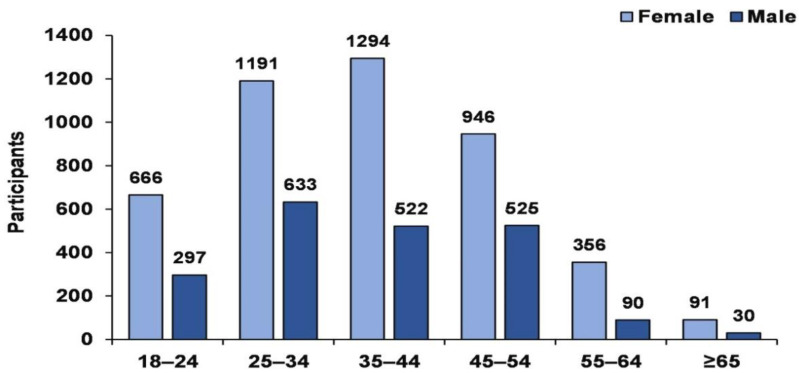
Research participant sample sizes by age groups by gender.

**Figure 2 ijerph-18-12809-f002:**
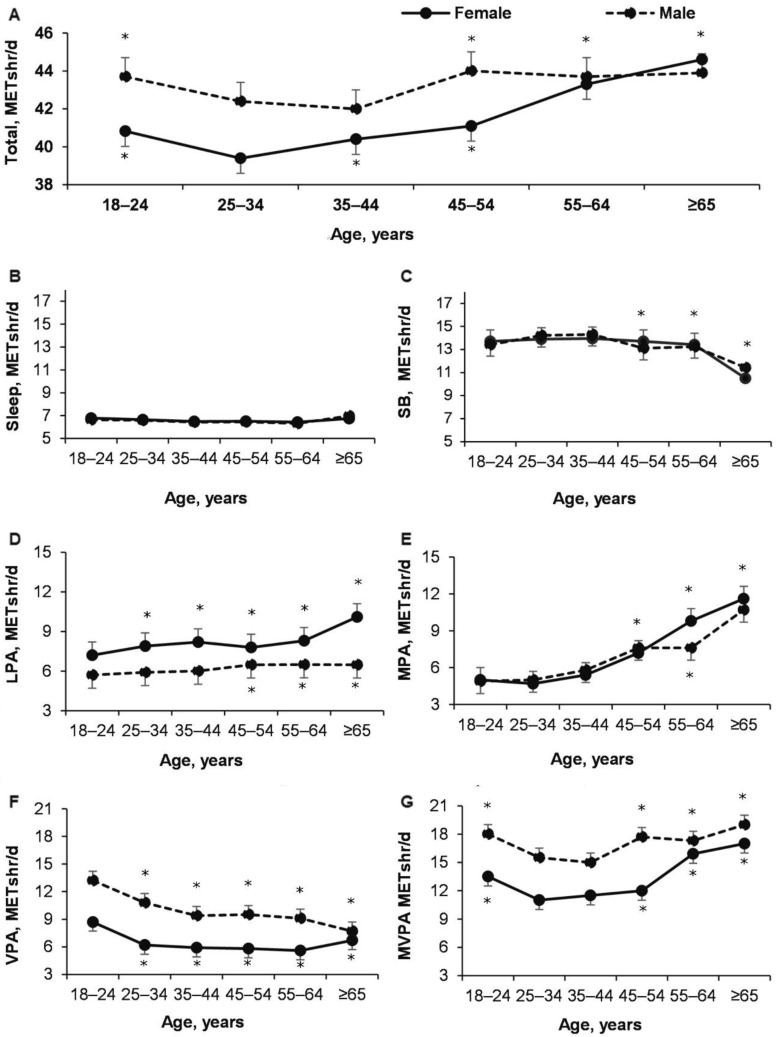
Changes during age total (**A**), sleep (**B**), SB (**C**), LPA (**D**), MPA (**E**), VPA (**F**), and MVPA (**G**) METs per day. (**A**). *—*p* < 0.05 compared to male 25–44 years and females compared to 25–34 years; (**C**). *—*p* < 0.05 compared to 18–44 years; (**D**). *—*p* < 0.05 compared to male 18–44 years and female with 18–24 years; (**E**). *—*p* < 0.05 compared to 18–44 years; (**F**). *—*p* < 0.05 compared to18–24 years; (**G**). *—*p* < 0.05 compared to 25–44 years. Note. Here in after: SB—sedentary behavior; LPA—light intensity physical activity; MPA—moderate intensity physical activity; VPA—vigorous intensity physical activity; MVPA—moderate and vigorous intensity physical activity.

**Figure 3 ijerph-18-12809-f003:**
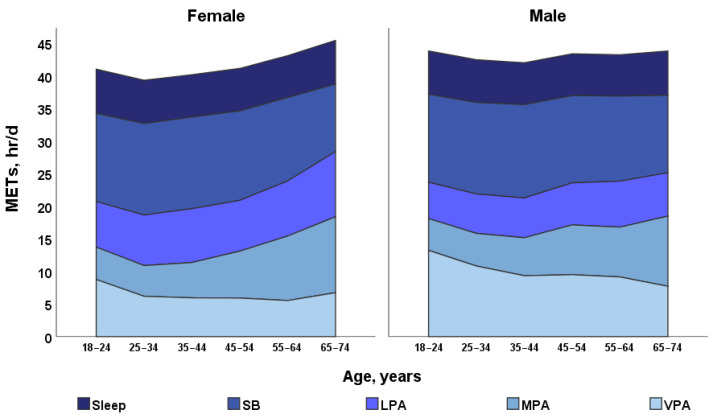
Sleep, SB, LPA, MPA, and MVPA changes in METs about gender and age differences.

**Figure 4 ijerph-18-12809-f004:**
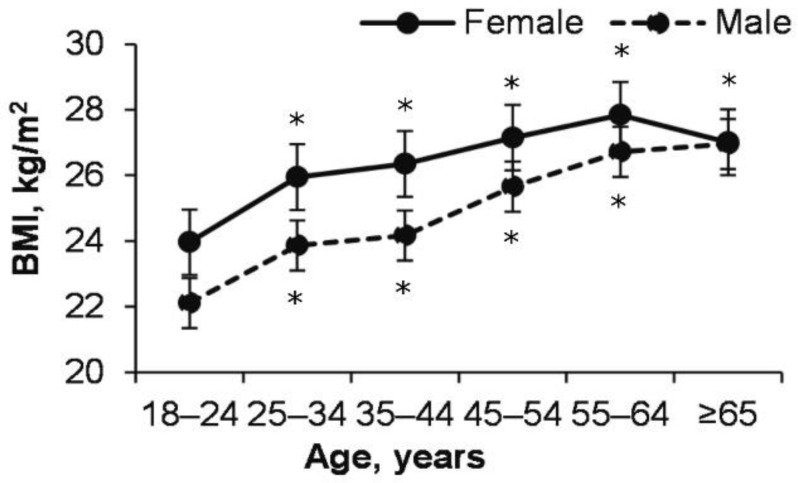
BMI changes during age. * *p* < 0.05 compared to 18–24 years.

**Figure 5 ijerph-18-12809-f005:**
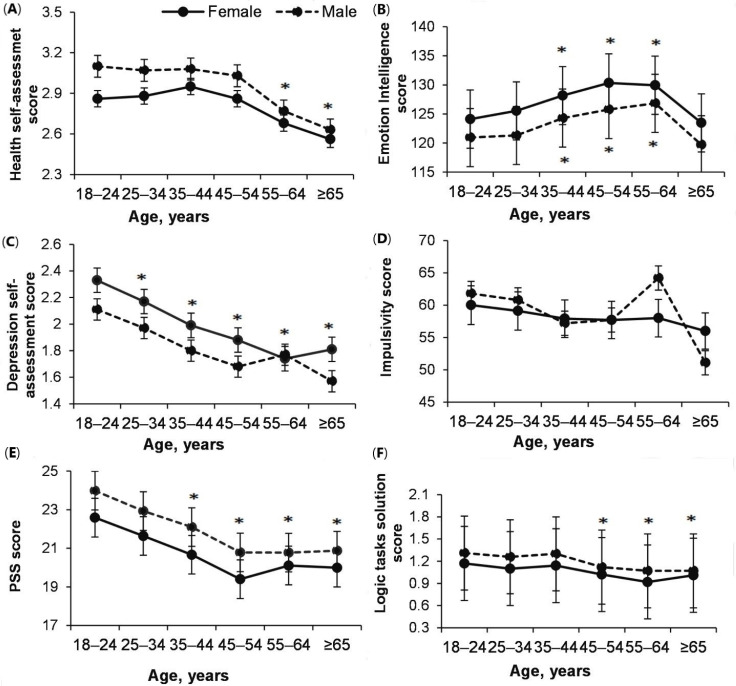
Changes in health (**A**), emotional intelligence (EI) (**B**), depression (**C**), impulsivity (BIS) (**D**), perceived stress (**E**) and reflexive thinking (logic task solution) (**F**) within age. (**A**). *—*p* < 0.05, compared to 18–24, 25–34, 35–44; 45–54 years; (**B**). *—*p* < 0.05, compared to 18–24, 25–34 and 65–74 years; (**C**). *—*p* < 0.05, compared to 18–24 years; (**E**). *—*p* < 0.05, compared to 18–24, 25–34 years; (**F**). *—*p* < 0.05, compared to 18–24, 25–34, 35–45 years.

**Figure 6 ijerph-18-12809-f006:**
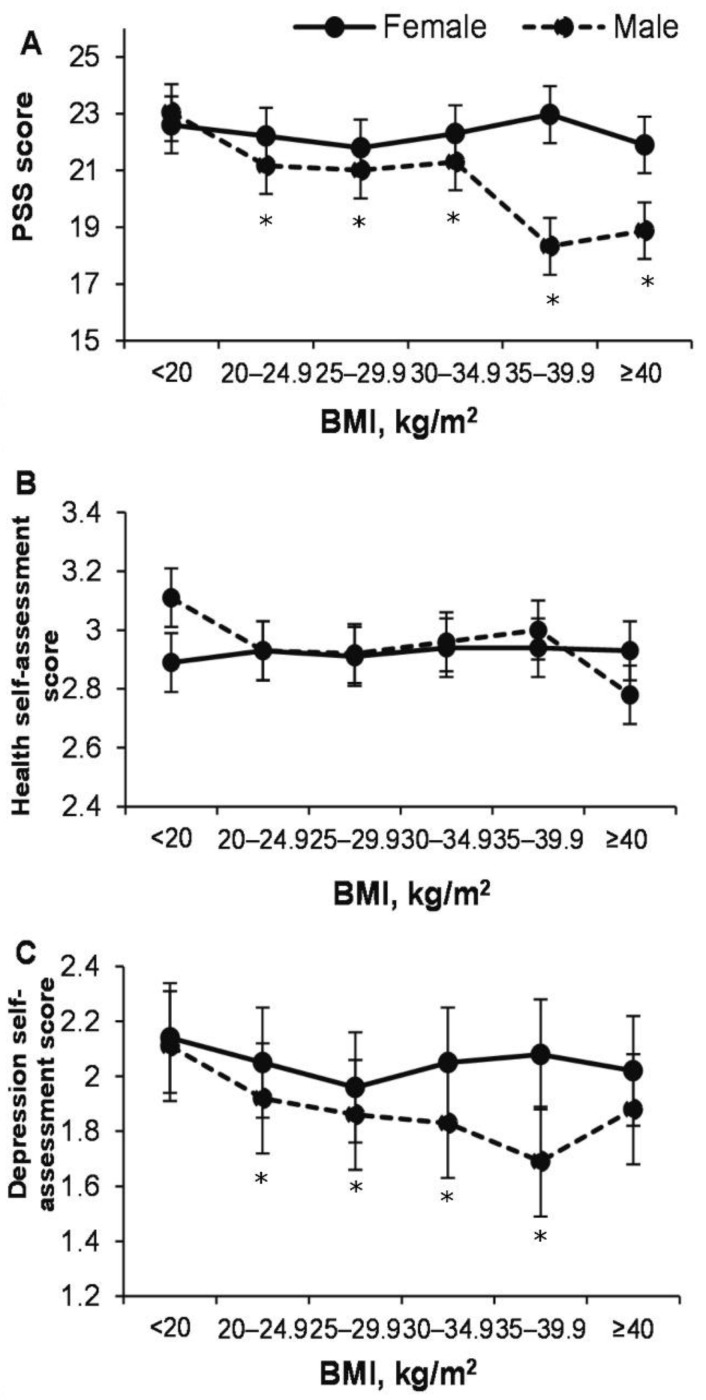
Relationship between BMI (kg/m^2^) and perceived stress (**A**), health (**B**), and depression (**C**). (**A**,**C**). * *p* < 0.05, compared to <20 BMI.

**Figure 7 ijerph-18-12809-f007:**
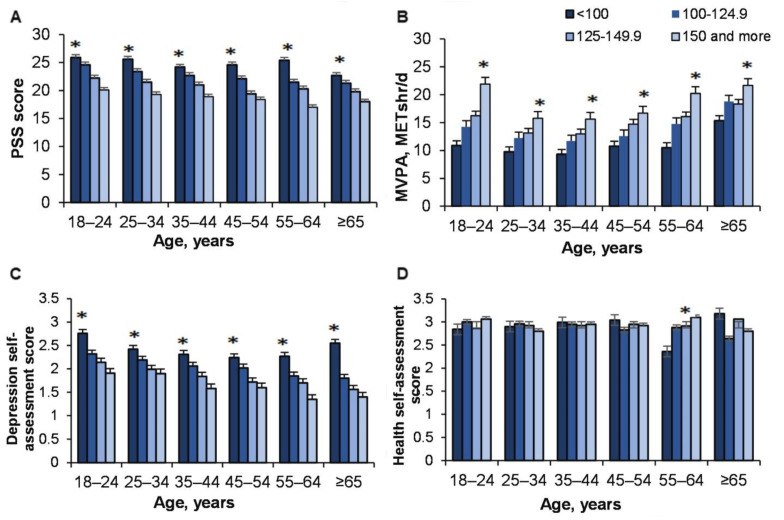
The effect of age and emotional intelligence (EI) on perceived stress (**A**), moderate and vigorous intensity PA (MVPA) (**B**), depression (**C**), and health (**D**). (**A**). *—*p* < 0.05 compared to EI 100–150 and more; (**B**). *—*p* < 0.05 compared to EI < 150; (**C**). *—*p* < 0.05 compared to EI 100–150 and more; (**D**). *—*p* < 0.05 compared to EI < 150.

**Figure 8 ijerph-18-12809-f008:**
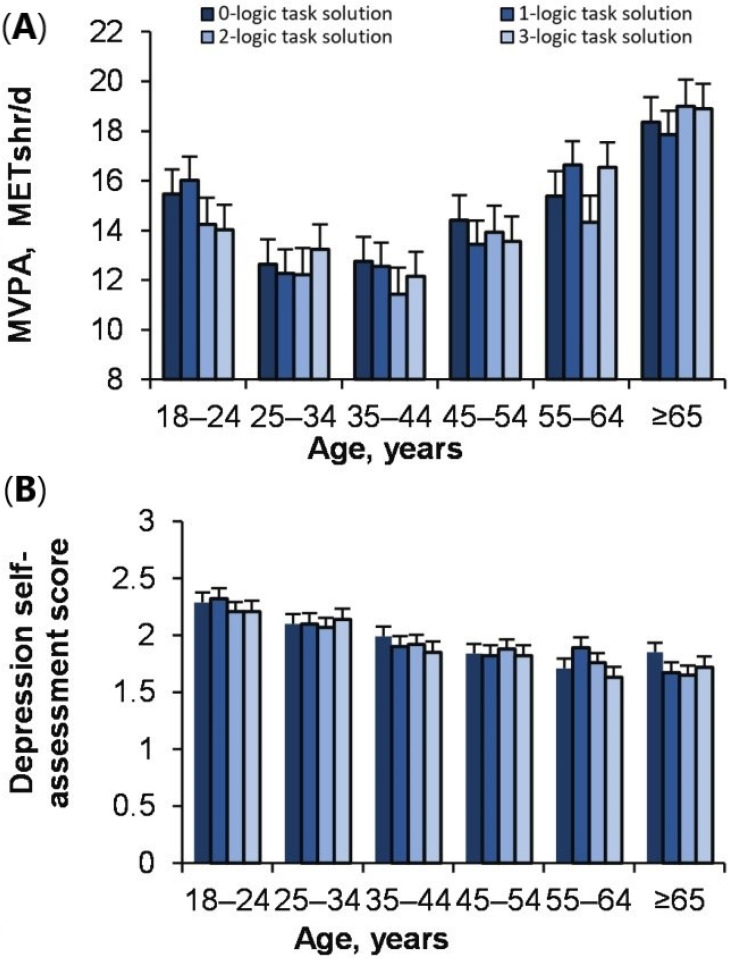
The age effect of logic task solution with MVPA (**A**) and depression (**B**).

**Table 1 ijerph-18-12809-t001:** Descriptive average values in female and male: health, logical task solution, impulsivity (BIS), EI.

		Gender				Chi-Square	*p*-Value
		Female		Male			
		*N*	*%*	*N*	*%*		
What is your health condition for the last year?	Excellent	760_a_	16.7%	474_b_	26.0%	95.53	0.000
	Good	2563_a_	56.4%	1010_a_	55.4%		
	Satisfactory	1111_a_	24.4%	301_b_	16.5%		
	Poor	112_a_	2.5%	39_a_	2.1%		
Logic task solution	None	2000	44	683	37.4	30.90	0.000
	One	1025	22.6	410	22.5		
	Two	715	15.7	327	17.9		
	Three	805	17.7	404	22.1		
Impulsivity, BIS	<50	908	20	212	14.1	2.10	0.54
	50–59	1875	41.3	906	52.2		
	60–69	1336	29.4	552	32.8		
	>70	426	9.4	154	10.9		
Emotional intelligence	<100	245	5.4	135	7.4	88.80	0.000
	100–124	1692	37.2	865	47.4		
	125–150	2324	51.1	763	41.8		
	>150	284	6.3	61	3.3		

Note: Values in the same row and sub-table not sharing the same subscript are significantly different at *p* < 0.05 in the two-sided test of equality for column proportions. Cells with no subscript (a,b) are not included in the test. Tests assume equal variances. Tests are adjusted for all pairwise comparisons within a row of each innermost sub-table using the Bonferroni correction.

**Table 2 ijerph-18-12809-t002:** Gender differences about participation in sports.

Gender						Age	Years		
			18−24	25−34	35−44	45−54	55−64	65−74	Total
Female	I do not exercise	Count	207_a_	473_b_	485_a, b_	377_b_	152_b_	32_a,b_	1726
		%	31.1%	39.7%	37.5%	39.9%	42.7%	35.2%	38.0%
	I am in a professional sport	Count	66_a_	46_b_	32_b,c_	8_c_	1_c_	1_a,b,c_	154
		%	9.9%	3.9%	2.5%	0.8%	0.3%	1.1%	3.4%
	I exercise by myself	Count	239_a_	337_b_	335_b_	272_b_	116_a,b_	28_a,b_	1327
		%	35.9%	28.3%	25.9%	28.8%	32.6%	30.8%	29.2%
	I exercise in a gym/health center	Count	154_a_	335_a,b_	443_c_	289_b,c_	87_a,b_	30_a,b,c_	1338
		%	23.1%	28.1%	34.2%	30.5%	24.4%	33.0%	29.4%
Male	I do not exercise	Count	46_a_	128_a,b_	123_a,b_	60_a,b_	25_a,b_	12_b_	394
		%	15.5%	20.2%	23.6%	23.8%	27.8%	40.0%	21.6%
	I am in a professional sport	Count	56_a_	54_b_	19_c_	7_c_	2_b,c_	1_a,b,c_	139
		%	18.9%	8.5%	3.6%	2.8%	2.2%	3.3%	7.6%
	I exercise by myself	Count	146_a_	283_a_	254_a_	134_a_	44_a_	14_a_	875
		%	49.2%	44.7%	48.7%	53.2%	48.9%	46.7%	48.0%
	I exercise in a gym/health center	Count	49_a_	168_b_	126_a,b_	51_a,b_	19_a,b_	3_a,b_	416
		%	16.5%	26.5%	24.1%	20.2%	21.1%	10.0%	22.8%

Each subscript letter (a, b, and c) denotes a subset of one category whose column proportions do not differ significantly from each other at the 0.05 level.

**Table 3 ijerph-18-12809-t003:** BMI changes within an age.

	BMI, kg/m^2^					Age	Years	
			18–24	25–34	35–44	45–54	55–64	65–74
Female	<20	Count	180_a_	250_b_	185_c_	61_d_	11_e_	1_e_
		%	27.0%	21.0%	14.3%	6.4%	3.1%	1.1%
	20–24.9	Count	389_a_	675_a,b_	685_b_	435_c_	131_d_	36_c,d_
		%	58.4%	56.7%	52.9%	46.0%	36.8%	39.6%
	25–29.9	Count	73_a_	194_b_	291_c_	293_d_	143_e_	38_e_
		%	11.0%	16.3%	22.5%	31.0%	40.2%	41.8%
	30–34.9	Count	16_a_	56_b_	102_c_	113_d_	53_d_	7_b,c,d_
		%	2.4%	4.7%	7.9%	11.9%	14.9%	7.7%
	35–39.9	Count	6_a_	14_a_	20_a,b_	26_c_	11_b,c_	7_d_
		%	0.9%	1.2%	1.5%	2.7%	3.1%	7.7%
	40 and more	Count	2_a,b_	2_b_	11_a,c_	18_d_	7_c,d_	2_c,d_
		%	0.3%	0.2%	0.9%	1.9%	2.0%	2.2%
Male	<20	Count	26_a_	11_b_	6_b_	1_b_	2_b_	0_a,b_
		%	8.8%	1.7%	1.1%	0.4%	2.2%	0.0%
	20–24.9	Count	185_a_	293_b_	194_c_	72_d_	22_d_	10_b,c,d_
		%	62.3%	46.3%	37.2%	28.6%	24.4%	33.3%
	25–29.9	Count	71_a_	269_b_	264_c_	139_c_	43_b,c_	17_b,c_
		%	23.9%	42.5%	50.6%	55.2%	47.8%	56.7%
	30–34.9	Count	12_a_	52_b_	46_b_	31_b,c_	18_c_	1_a,b_
		%	4.0%	8.2%	8.8%	12.3%	20.0%	3.3%
	35–39.9	Count	3_a_	8_a_	9_a_	7_a_	2_a_	1_a_
		%	1.0%	1.3%	1.7%	2.8%	2.2%	3.3%
	40 and more	Count	0_a,b_	0_b_	3_a,b,c_	2_a,c,d_	3_d_	1_c,d_
		%	0.0%	0.0%	0.6%	0.8%	3.3%	3.3%

Each subscript letter (a, b, c, d, and e) denotes a subset of one category whose column proportions do not differ significantly from each other at the 0.05 level.

## Data Availability

The data presented in this study are available on request from the corresponding author.

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
