# Peer review of "Physical Activity, Stress, Depression, Emotional Intelligence, Logical Thinking, and Overall Health in a Large Lithuanian from October 2019 to June 2020: Age and Gender Differences Adult Sample"

_ijerph, 2021, doi:10.3390/ijerph182312809_

Round 1
Reviewer 1 Report
I would like to thank the authors for addressing my concerns. I have found the manuscript as publishable, however, I suggest authors reread the manuscript and revise some long sentences which are hard to understand. Such as:
In general, our primary hypothesis was only partially confirmed than, as we expected, overweight and obesity in young adults — from 18 to 34 years — increased and especially in the male, and at the same time the life-span interval decreased in MVPA (consistent with data from other researchers, [11], but contrary to expectations, subjective health assessment did not change, and feelings of stress and depression decreased (for impulsiveness did not change with age).
Author Response
Dear Reviewer,
Thank you for the opportunity to revise and resubmit the manuscript, “Weight and obesity in young adults extremely increased, but the expression of stress and depression decreased, the health assessment did not change: Why?” for consideration in the International Journal of Environmental Research and Public Health.
We have submitted a revised version of the manuscript that addresses reviewers comments. In this letter, we address each of the points raised and point to the specific change, or in rare cases, our reason for not making the change. The manuscript resubmitted has changes highlighted, as requested. We appreciate the effort reviewers that we believe will strengthen this manuscript.
Thank you.

Reviewer 2 Report
I want to thank the authors for the changes made.
I propose not to use the personal form, but the impersonal one, e.g. instead of "we have tested", it has been "tested".
The work still needs to be corrected.
Lines: 31, 87 - please explain the abbreviation "FA".
Lines: 53-54 - please complete that it is about the negative impact.
The work contains many stylistic errors:
Line 115 - remove the dot after [28].
Line 129 - If the sentence begins with a capital letter, a dot is missing.
Line 146 - End the sentence with a dot, not a comma.
Line 279 - Unify the spelling of "male's" and "Female's". Why does the text use lowercase letters and sometimes uppercase letters (Female, female, male, Male)? Please see e.g. lines 280, 282, 285 and 291, 301, 346, 350.
Errors on lines 308-310.
Lines 314-316 unclear sentence. Please rewrite.
Author Response
Dear Reviewer,
Thank you for the opportunity to revise and resubmit the manuscript, “Weight and obesity in young adults extremely increased, but the expression of stress and depression decreased, the health assessment did not change: Why?” for consideration in the International Journal of Environmental Research and Public Health.
We have submitted a revised version of the manuscript that addresses reviewers comments. In this letter, we address each of the points raised and point to the specific change, or in rare cases, our reason for not making the change. The manuscript resubmitted has changes highlighted, as requested. We appreciate the effort reviewers that we believe will strengthen this manuscript.
Thank you.

This manuscript is a resubmission of an earlier submission. The following is a list of the peer review reports and author responses from that submission.
Round 1
Reviewer 1 Report
The work contains a lot of errors and requires thorough improvement.
The main considerations are as follows:
Abstract:
“Young adults (18 to 34 years) and male” - unclear. Please rewrite.
"FA", "EI", "MVPA" - do not use abbreviations without first explaining the concept.
Introduction: The topic of work is current and essential. Admission is definitely too poor. Supplement with references.
Consider, among other things, the impact of the pandemic on overweight and obesity in the analyzed group.
Materials and Methods:
Have all the questionnaires been completed correctly? It is impossible.
Would you mind indicating how many questionnaires were completed (including those completed correctly)?
Point 2.1. I point 2.2. The same text. “We used an online survey to collect information through https://docs.google.com/forms/. All participants completed the online questionnaires. An online survey using the Google Forms platform was distributed by researchers through social media (Facebook) and personal contacts (WhatsApp) ".
2.3. Measures
“We calculated how much energy (METs)”. Earlier, an abbreviation appears in the text without its explanation. Punctuation errors appear. It is difficult to indicate them because there is no line numbering.
For the fragments: “Danish Physical Activity Questionnaire (DPAQ)” and “Subjective health assessment. A four-point scale was used for this: poor health (1 point); satisfactory (2 points), good (3 points); excellent (4 points) ” please add a reference.
"Five-point scale ranging from 0 to 4". Please check.
“Higher scores indicate higher levels of perceived stress. In the HADS [25], participants were asked to answer 14 questions (i.e., 7 questions each for anxiety and depression subscales) about feelings at that moment on a 4-point scale ranging from 0 to 3 ” - unclear fragment. Please rewrite.
2.4. Data Analysis.
Describe the statistical analysis for specific analyzes/tests in detail.
The text contains statistical analyzes that are not described in the "Data Analysis" section.
3. Results.
Table 1. The question asked does not apply to reality (taking into account covid-19) "What is your health condition for the last year?"
Figure 3. “Sleep, SB, LPA. MPA, MVPA changes in METs about gender and age differences ". Do not use abbreviations in the chart name.
Please, Put explanations of abbreviations under the diagrams.
The text under Figure 3 should be redrafted as it will be difficult for the reader to understand the data.
4. Discussion
The same fragment appears in work in 3 different places. “Two hypotheses were formulated before the study. The first hypothesis: overweight and obesity in young adults - from 18 to 34 years - and males, in particular, should increase sharply and this should be associated with decreased FA, abruptly deteriorating subject health, increased stress, depression, and poorer emotion management / emotional intelligence. Second hypothesis: since human choices are not necessarily influenced by a rational / reflective mind (explicit knowledge)"
Reviewer 2 Report
This manuscript aims to prove the hypothesis that overweight and obesity in
young adults are increasing due to decreasing physical activity and increased stress, depression, and poorer emotion management and emotional intelligence. I have major concerns related to the manuscript and listed some of them below for the authors' reference:
- There is no logical flow in the introduction. The authors started with physical activity and then surprisingly continued with obesity / BMI with one sentence in the second paragraph. And then again continued with physical activity and its effect on chronic diseases.
- The authors did not explain what this manuscript will bring to existing literature. Their rationale has not been stressed enough.
- It is written that authors reached to participants via WhatsApp (personal contacts) which consequently introduced the selection bias.
- There is serious repetitiveness throughout the manuscript.
- The authors should explain the methods in detail.
- It is hard to follow the results section. The sentences are similar to each other. I felt that I am reading the same sentence over and over again.
- In the way the authors presented the results, it seems that it is a longitudinal study rather than a cross-sectional one. For example, the authors wrote that "emotional intelligence increased steadily with age (18-24 to 65-74 years)" which sounds like participants who were 18-24 years old had higher emotional intelligence when they aged 65-74 years. Instead, authors should write a sentence such as 'older participants had higher emotional intelligence than younger adults.'
- There are problems with abbreviations - either the authors did not explain them or wrote some of them wrong such as FA (I assume it is physical activity?)
- In conclusion, I suggest the authors rewrite the manuscript in a more logical way.